# Device for Contact Measurement of Turbine Blade Geometry in Robotic Grinding Process

**DOI:** 10.3390/s20247053

**Published:** 2020-12-09

**Authors:** Dariusz Szybicki, Andrzej Burghardt, Krzysztof Kurc, Piotr Gierlak

**Affiliations:** Department of Applied Mechanics and Robotics, Faculty of Mechanical Engineering and Aeronautics, Rzeszow University of Technology, 35-959 Rzeszów, Poland; dszybicki@prz.edu.pl (D.S.); andrzejb@prz.edu.pl (A.B.); kkurc@prz.edu.pl (K.K.)

**Keywords:** blade measurement, measuring device design, accuracy, repeatability

## Abstract

The article discusses the design, implementation, and testing of the accuracy of a measuring device used to measure the thickness of aircraft engine blades subjected to a robotic grinding process. The assumptions that the measuring device should meet were presented. The manufactured device was subjected to accuracy and repeatability tests using a standard workpiece. The analysis of research results proved that the measuring device exhibits an accuracy of one order of magnitude better than the accuracy required for blades. For control of the grinding process, the results should be perceived as appropriate. Then, the device was subjected to verification consisting in using it to measure the thickness of aircraft engine blades. The constructed device can be used, not only for inspection of final products, but also for control of the robotic grinding process because thanks to the output interface it can be used in the robotic station’s feedback loop.

## 1. Introduction

Turbine and turbine engine compressor blades are key elements in structures of this type. Such systems are commonly used, e.g., in turbine generators in power plants, or aviation drives [1]. Blade quality largely influences the efficiency, durability, and reliability of these devices. Therefore, the blade manufacturing process is strictly controlled to ensure its repeatability, and the blades’ geometric properties are subject to rigorous assessment [2]. Similar control is performed during the blade repair process, which extends the life of the system and increases its economic efficiency [3,4]. An important aspect in choosing the method for geometric parameter control is the fact that the final shape of the blade depends on the final grinding operation. Apart from shaping, grinding helps to achieve appropriate surface smoothness.

Due to shape complexity, the measurement of the blade geometrical parameters constitutes a difficult process. The blade airfoil is a curvilinear three-dimensional surface, sometimes with considerable twisting (Figure 1a). Various solutions are used to measure the geometric parameters of the blades subjected to grinding. They can be divided into contact and contactless methods [5]. The contactless methods mainly include laser and optical scanning, as well as computer tomography. 2D or 3D scanners are used for laser and optical scanning. The advantages of these solutions include short measurement time, a large amount of data obtained during one measurement, and relatively low cost of data acquisition and processing. Some laser measuring devices have an accuracy similar to tactile solutions, but the disadvantage of laser methods lies in degraded accuracy when measuring glistening surfaces. In turn, the disadvantage of optical scanning lies in the high sensitivity of most of the scanning devices to the type of lighting, dust, or reflections on the surface. In addition, if the measurements necessitate combining fragments of the scanned surface, as is the case with the scanning of large objects or objects with a complex shape, the accuracy of the measurement is lower compared to a single-surface image. Computer tomography is another contactless method, but due to the high cost of measurement and data processing, it is usually used only in cases where, apart from the geometric parameters of the blade surface, knowledge of its internal structure is also required. The measurement and processing time is significantly longer than for scanning. The basic contact method used to measure the geometric parameters of the blades consists of using coordinate measuring machines (CMM) equipped with measuring probes. This method is characterized by top accuracy, which unfortunately results in the longest measurement and data processing time. In addition, both CMM measurement time and cost increase dramatically as a function of the complexity of the object’s shape, significantly more than in other measurement methods. Therefore, simpler dedicated instruments are sometimes used for contact measurements. A detailed comparison of the blade measurement methods can be found in the papers [5,6,7].

The selection of the type of measurement method is largely determined by the manufacturing tolerance of the blades, which is usually related to their size [8]. Large blades usually have a large manufacturing tolerance and scanning can be used to assess their geometric parameters. On the contrary, small blades usually exhibit narrow manufacturing tolerances, which forces the use of precise measurement methods, i.e., contact methods. The manufacturing tolerances are sometimes so small that it is necessary to grind the blade surface only once, which makes material losses so small that it is difficult to detect them when scanning. The issues with measuring small blades are discussed later in this paper.

To design the measuring device correctly, it is necessary to take into account the character of the blade grinding process. This process is very often carried out either manually or with the help of robotic stations. There are also solutions using CNC machines. However, their application is largely problematic due to significant machine stiffness and low stiffness of the blade, which requires the use of an airfoil support system for the blades. Another problem lies in the fact that the most efficient blade-making processes, electro-drilling [9] or casting [10,11], are characterized by low repeatability and the airfoil position in relation to the dovetail is highly variable (Figure 1b). This is due to the low stability of the processes aimed at giving the blade its approximate shape, e.g., by casting or electro-drilling. It is very important to note that the strict requirements apply to blade thickness, while a relatively large deviation in airfoil position relative to the dovetail is allowed. This is due to the fact that dovetail is usually processed only after airfoil, which provides large correction possibilities for the mutual position of these elements at this stage. Due to the large spread of the airfoil position in relation to the dovetail, the manual blade processing method has become very popular; however, it is associated with a number of problems with repeatability and ensuring occupational safety and health. The use of robotic stations [12,13] present a solution to this problem, due to the fact that such stations are more flexible than CNC machines. This applies both to flexibility in terms of physical characteristics and in terms of software capabilities.

The precision of industrial robots is important for the robotic grinding process. It is determined by two parameters: accuracy and repeatability. The robot’s accuracy determines how close the robot can get to a given point in the working space. Repeatability determines how close a robot can get to a point previously reached. Industrial robots provide quite good repeatability, defined according to ISO 9283 [14] at a level as low as about 0.03 mm, but this can deteriorate more than twice in linear paths. Unfortunately, in the process of robotic blade grinding, good repeatability of industrial robots does not provide that much of an advantage because the blades themselves are characterized by low repeatability, and an industrial robot cannot process different blades with the same programmed path. Contrary to popular belief, industrial robots are not very accurate. This is due both to errors in the execution of kinematic units and to errors in the solutions of kinematic equations used by the robot axis controller. The accuracy of an industrial robot can range from a few to even a dozen millimeters. To improve accuracy, manufacturers of industrial robots offer added software that improves the accuracy (in accordance with ISO 9283 [14]) to about 0.3–0.7 mm. Improved accuracy is achieved by calibrating a specific mechanical unit. The costs of such additions are similar to the cost of the robot itself. However, even this accuracy is too low, as the accuracy required for blades is in the order of 0.1 mm.

The problem of low blade repeatability can be solved in two ways. The first way is to use flexible tools that would allow the machining system to automatically adapt to the blade. Belt grinders are the most popular solution in this area [12,13]. However, the tensioning system, vibrations, rapid wear, and the need to change the belt present numerous problems, which make it difficult to fully automate the process. Another method is the use of industrial robots with force control, which realize movement on the ground surface with simultaneous control of the pressing force, which is one of the variables of the grinding process [15]. In this case, instead of setting the exact path to follow the direction normal to the workpiece surface, the desired tool’s pressing force is set to the blade itself, which ensures contact between the tool and the workpiece [16,17]. This avoids the problem of both low accuracy of industrial robots and low repeatability of the blades.

This paper presents a device which is designed to carry out measurements in the process of grinding small blades with control over the grinding tool’s pressing force. Due to the small size and narrow tolerances of the blades, the presented device uses a contact measurement method. Section 2 presents the measuring device and the criteria it should meet. Section 3 presents the results of the accuracy and repeatability analysis, whereas Section 4 presents the results of aircraft engine blade measurements. The paper ends with a discussion of the results.

## 2. Measuring Device

### 2.1. Assumptions for the Measuring Device

The measuring device is to be used to control blade dimensions in the robotic grinding process. For this reason, it must have a communication interface in a standard compatible with the robot controller used for the process. The measurement of the blade’s geometrical parameters and grinding will be carried out cyclically until the appropriate blade shape is achieved, as shown in Figure 2. In the subsequent stages of the grinding process, process parameters are corrected based on the results of the blade geometry measurements.

From a metrological point of view, an important aspect of the process is the method of attachment of the blade. During the measurement, the robot places the blade in the measuring space of the device. The blade is attached to the robot’s arm with a gripper whose jaws clamp on the dovetail. The processing of the blade and the measurement of its geometric parameters are carried out in an automatic cycle without changing the method of fixing. This approach eliminates the problem of fastening repeatability.

The required accuracy of the measuring device is determined by the required accuracy of the airfoil. The required tolerance for small airfoils is usually about ±0.1 mm. The accuracy of the measuring device should be at least an order of magnitude higher, i.e., at least 0.01 mm. The robot’s accuracy is not important here, because, in the process of controlling the robot’s movement with force control, the shape of the airfoil does not result from the exact execution of the robot path, but from the pressure of the tool with a given force from excess material.

It follows from the discussion that the blade thickness measuring device should meet three key requirements:Have an accuracy of at least 0.01 mm;Exhibit repeatability of not less than 10% of the airfoil’s make tolerance;Provide feedback on the blade condition during the grinding process.

### 2.2. Description of the Measuring Device

The measuring device (Figure 3) consists of eight contact measuring elements, a housing, and a communication interface (Figure 4). GT2-A32 sensors (Keyence, Osaka, Japan) were used as measuring elements. The sensors are arranged into pairs and placed opposite each other so that the thickness of the blade airfoil at a selected point could be determined with pairs of sensors. The sensors are equipped with a gas spring controlling the movement of the measuring element. Therefore, the heads of the contact sensors require a supply of compressed air. For this purpose, a valve island controlled by a robot via ProfiBus (Figure 4) is used. At zero pressure, the measuring probe is retracted and the maximum extension of 32 mm occurs at 0.5 Mpa. The manufacturer’s declared sensors’ accuracy is 0.003 mm at a resolution of 0.0005 mm. The use of eight sensors, and their arrangement, results from the number and arrangement of measuring points for typical blades. The device is designed to allow us to determine the blade’s airfoil thickness at three points and its width for any number of blade cross-sections. The S_1_ and S_2_ sensors are equipped with contact plates instead of rod probes which ensures reliable contact between the sensor and the side edge of the blade’s airfoil. The arrangement of the sensors are shown in Figure 5. The device casing was made with the accuracy of the location of the mounting holes for contact sensors of 0.005 mm.

### 2.3. Blade Measurement Algorithm

The algorithm used for measuring airfoil thickness is shown as a block diagram in Figure 6. Due to the relatively large size of the measuring device housing, its temperature expansion has a significant impact on the distances between sensor pairs. To eliminate the influence of this phenomenon on the measurement results, each time before airfoil measurement a two-stage calibration procedure is carried out. In the first stage, the sensors are calibrated in the following pairs: S_3_ and S_4_, S_5_ and S_6_, and S_7_ and S_8_. Calibration of these sensors consists of extending their contact probes until the upper sensors probes come into contact with the lower ones and recording the sensors’ measurement results as c_3_, c_4_, c_5_, c_6_, c_7_, and c_8_, respectively. The second calibration step concerns the pair of S_1_ and S_2_ sensors. These sensors are too far away for their contact probes to come into contact with each other. At maximum extension, the distance between the probes is 23.482^±0.001^ mm. Therefore, a reference element, made as an integral part of the robot’s gripper used for gripping the blade during the machining process, is used for the calibration procedure. Surfaces of the model element have been ground and its width has been determined with CMM and is n = 49.100^±0.001^ mm. The gripper together with the machined blade is placed in the measuring space of the device so that it is between the S_1_ and S_2_ sensors. Then, the sensors’ contact probes are extended until they come into contact with the surfaces of the reference element. The measurement results from the sensors are recorded as c_1_ and c_2_, respectively. The blade is then placed in the measuring space of the device in successive positions to determine airfoil width and thickness in N selected cross-sections. The contact probes of all sensors are extended until the airfoil comes into contact with the blade in each cross-section. The sensors’ measurement results are recorded as m_1i_, m_2i_, m_3i_, m_4i_, m_5i_, m_6i_, m_7i,_ and m_8i_, respectively, where i is the cross-section number and i = 1, 2, …, N. The width of the airfoil is then determined for each cross-section from the following equation:w_i_ = n + c_1_ + c_2_ − (m_1i_ + m_2i_),(1)
and airfoil thickness in each cross-section is determined in three points from these equations:t_1i_ = c_3_ + c_4_ − (m_3i_ + m_4i_),(2)
t_2i_ = c_5_ + c_6_ − (m_5i_ + m_6i_),(3)
t_3i_ = c_7_ + c_8_ − (m_7i_ + m_8i_).(4)

Please note that the zero position of each sensor probe is at its maximum extension, and the c_i_ variables take zero as their value then. At incomplete extension, the c_i_ values indicate the distance between the probes and the zero positions.

## 3. Accuracy and Repeatability of the Measuring Device

The accuracy and repeatability test was carried out using a prismatic reference element (Figure 7). Accuracy is defined as the deviation of the mean value from the measurements from the actual value of the measured feature. In turn, repeatability is defined as variability of measurement results obtained by a given operator when measuring the same part many times under the same measurement conditions in a short time interval [18]. In this case, the superordinate device control system plays the role of the operator. As the operator does not change, analyze reproducibility was not analyzed. It only makes sense if measurements are carried out by more than one operator. The repeatability of fixing the blade in the robot’s gripper is not taken into account as it has no influence on the measurement result. This is because the thickness measurements are not referenced to any reference base. Furthermore, the repeatability of the fastening will not affect the measurement results during the blade grinding process as the grinding and measurement procedures are performed alternately but with one fastening.

For the purpose of analysis, the same reference element has been made ten times under the same conditions in short time intervals. The data obtained are shown in Table 1. The last row of the table shows the dimensions of the reference element obtained with CMM. Table 2 presents the results of the analysis.

Accuracy of the measuring system was determined from equation [18]:(5)$$A=\;\left|x_{a}-\bar{x}\right|,$$
where $x_{a}$ is the actual value of the measured characteristic (last row in Table 1), $\bar{x}$ is the mean value of the measured characteristic (first row in Table 2). The repeatability was determined using equation [18]:(6)$$\mathrm{EV}=5.15\frac{r}{d_{2}},$$
where r is the range of measured values of a given feature (the difference between the maximum and minimum measurement result for a given feature—the second line in Table 2), $d_{2}$ is a tabulated coefficient depending on the number of operators, measurement series, and measurements in the series [19]. For one operator, one series, and thirty measurements in the series it is $d_{2}=3.078$. For improved interpretation of the results, it is more advantageous to determine the relative repeatability using the formula [18]:(7)$$\%\mathrm{EV}=\frac{\mathrm{EV}}{T}100\%.$$

This is the reproducibility determined in relation to the tolerance range of the measured characteristic T. The tolerance range is assumed to be T=0.2 mm. This is due to the fact that the blade is to be made with a tolerance of ±0.1 mm, i.e., the tolerance range is 0.2 mm. It is assumed in [18] that %EV≤10% is an acceptable result, 10<%EV≤30% is a conditionally acceptable result, and %EV>30% is an unacceptable result.

The uncertainty of the measurement result was determined according to the ISO standard [20]. The experimental variance of the observations, which estimates the variance of the probability distribution of x, is given by
(8)$$s^{2}\left(x\right)=\frac{\sum_{j=1}^{n}{\left(x_{j}-\bar{x}\right)}^{2}}{n-1},$$
where $x_{j}$ is the j-th observation, n=30 is the number of observations in the series. The best estimate of the variance of the mean $\bar{x}$ is
(9)$$s^{2}\left(\bar{x}\right)=\frac{s^{2}\left(x\right)}{n}.$$

The experimental standard deviation of the mean is the positive square root of the expression (9):(10)$$s\left(\bar{x}\right)=\frac{\sqrt{s^{2}\left(x\right)}}{\sqrt{n}}.$$

Values (9) and (10) quantify how well $\bar{x}$ estimates the expectation of x, and that it may be used as a measure of the uncertainty of $\bar{x}$. From the interpretation point of view, it seems more advantageous to use the expression (10), called the type A standard uncertainty, because its unit is the same as the unit of the measured quantity. The values of the uncertainty determined in this way are presented in Table 2.

From the analysis of the results shown in Table 2, it follows that the accuracy of the measuring instrument with taking into account the uncertainty is no less than 0.008 mm and the relative repeatability is better than 10%. Both the achieved accuracy and the repeatability meet the requirements set out in Section 2.1, thus the measuring instrument has appropriate parameters and can be used to measure airfoil geometry.

## 4. Airfoil Geometry Measurement

The airfoil geometry in three cross-sections was measured using the algorithm presented in Section 2.3. Figure 8 shows the view of the device during airfoil width (Figure 8a) and thickness measurement at three points (Figure 8b). The individual dimensions are shown in Figure 9a. The results obtained are shown in Table 3. The total duration of the measurement procedure includes placing the blade in the measurement space, measuring the geometric parameters of the blade, and returning the robot arm to its starting position. The operation time of the sensors, consisting in extending the heads of sensors, stabilizing them, saving data, and returning the heads, is 0.5 s for a single measurement. Moreover, the procedure takes into account the time needed to damp vibrations of the robot arm and blade before each measurement. The entire procedure takes 16 s.

The measurement results presented in Table 3 still need corrections due to the airfoil curvature. This issue is presented in Figure 9b in detail using the t_1_ dimension as an example. The airfoil thickness g_1_ should be determined in the direction normal to the surface. Due to surface curvature, the measuring device determines dimension t_1_, which is not the thickness but the vertical distance between two points. There is angle *α*_1_ between the vertical axis and the direction normal to the surface, its value is known from the technical requirements. With the rough assumption that the OAB is a rectangular triangle with a right angle at vertex A, it is possible to write $g_{1}=t_{1}{\cos\alpha }_{1}$. Similarly, in order to determine the remaining thicknesses, it is possible to write $g_{2}=t_{2}{\cos\alpha }_{2}$ and $g_{3}=t_{3}{\cos\alpha }_{3}$. In each of the three measured points and each cross-section, the angles are different. The values of the angles, determined on the basis of the working drawing of the blade, are presented in Table 4. Corrected blade dimensions are shown in Table 5.

Moreover, the impact of possible changes in the orientation of the blade relative to the measuring device on the measurement results was investigated. If airfoil orientation is different than assumed, then the angles shown in Table 4 will be slightly different and the correction of the blade dimensions will be biased. There are three major reasons for a change in airfoil orientation:

Errors in the orientation of the last part of the robot’s arm not exceeding 0.0001 rad because the robot’s controller has a software add-on that improves positioning accuracy;Gripper’s blade clamping error with a gripper that is no more than 0.005 rad in the system under analysis;Allowable blade orientation error, relative to the dovetail, not exceeding 0.003 rad.

Taking into account all these inaccuracies, the maximum value of the orientation error was estimated at 0.0081 rad. The impact of this error on the corrected airfoil dimensions is calculated and presented in Table 5. It appears that the error in dimension correction should not exceed ±0.0019 mm.

## 5. Discussion

The paper presents the design and quality testing of a device for measuring airfoil geometry. What is the most important from the point of view of possibly using a device in the robotic process of grinding blades for controlling dimensions, is the possibility of cooperation with the robot controller, appropriate accuracy, and repeatability. Analysis of the measurement data shows if unknown correction errors resulting from blade orientation are taken into account, measurement accuracy for the described device would not be worse than 0.01 mm. The device’s relative repeatability meets the criteria specified for measuring devices and falls within the 10% tolerance range for the measured value.

The constructed device is designed for the measurement of small blades. However, it can easily be adapted to measure larger parts. It is enough to change the design of the housing or the location of the sensors. However, it should be kept in mind that the range of each of the measuring probes is small, a few dozen millimeters at most. This makes it impossible to build a universal device for measuring both small and large parts.

More work is needed to re-engineer the device and adapt it for use in industrial conditions. Among other things, in order to protect the measuring space from contamination, it will necessitate the use of a tight casing with the possibility to open up for measurements.

## Figures and Tables

**Figure 1 sensors-20-07053-f001:**
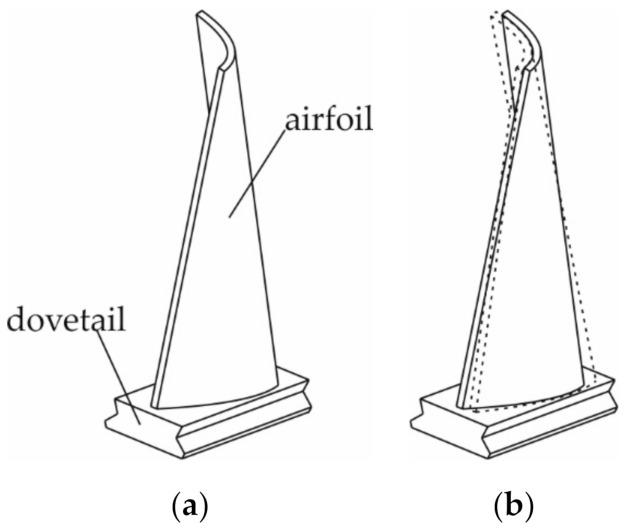
Blade: (**a**) main blade parts: airfoil and dovetail; (**b**) illustration of the problem with variability of the airfoil position in relation to dovetail.

**Figure 2 sensors-20-07053-f002:**
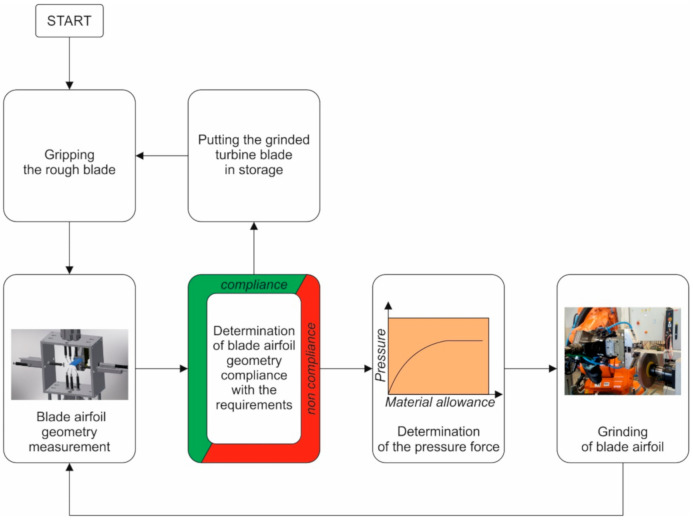
Schematic representation of the robotic grinding process.

**Figure 3 sensors-20-07053-f003:**
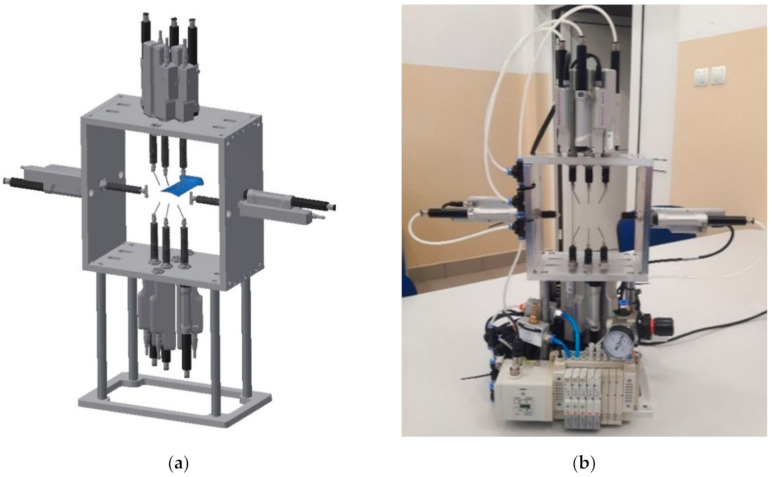
Device for contact airfoil measurement: (**a**) CAD model; (**b**) device projection.

**Figure 4 sensors-20-07053-f004:**
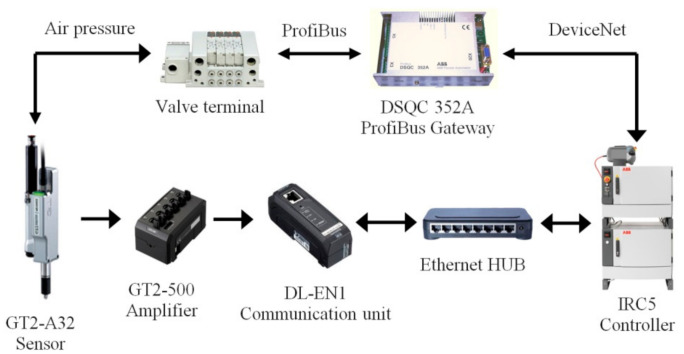
Connection of the contact sensor to the robot controller.

**Figure 5 sensors-20-07053-f005:**
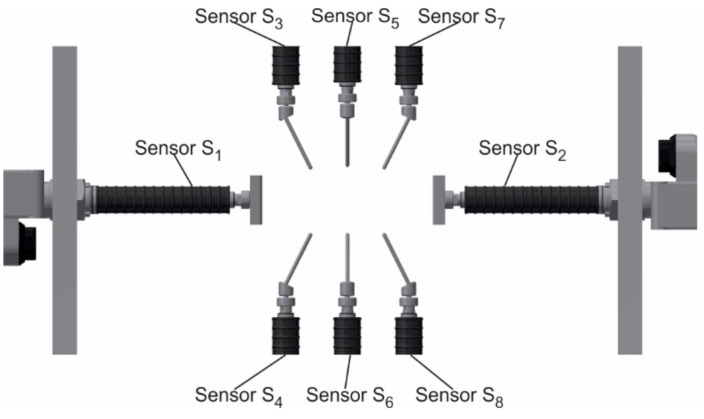
Arrangement of the contact sensors.

**Figure 6 sensors-20-07053-f006:**
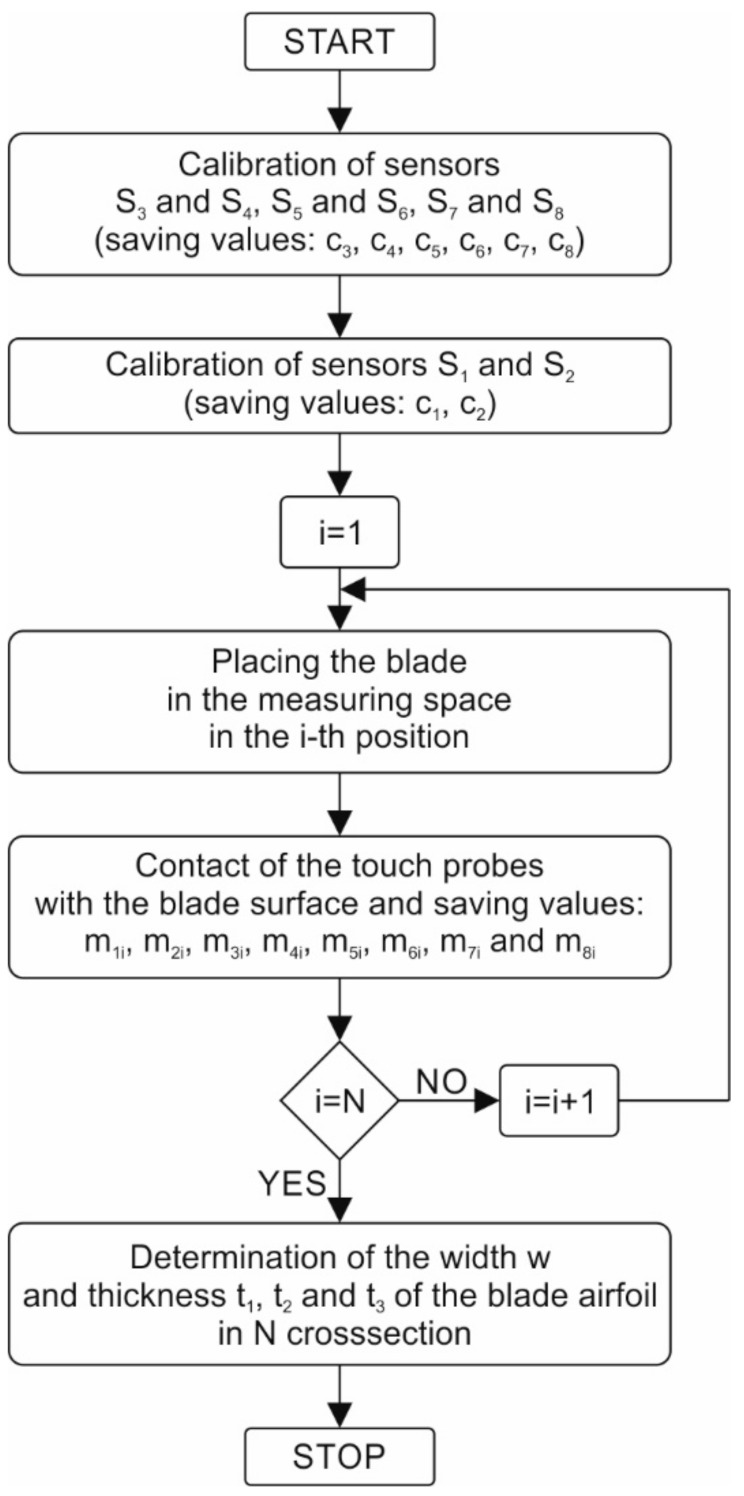
Block diagram of the measurement process.

**Figure 7 sensors-20-07053-f007:**
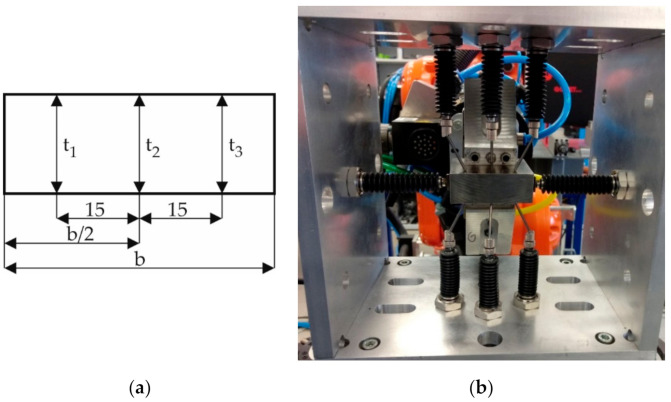
Testing accuracy and repeatability of the measuring device: (**a**) dimensions of the reference element; (**b**) device during measurement.

**Figure 8 sensors-20-07053-f008:**
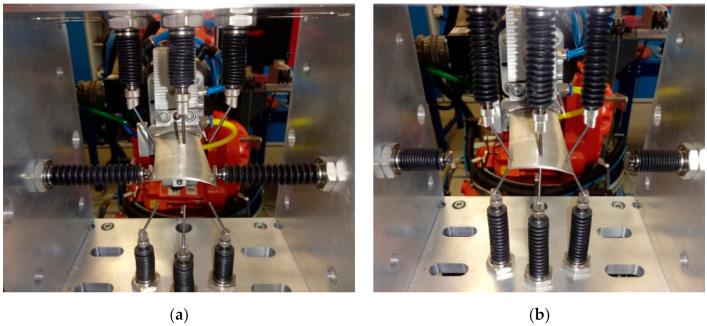
Airfoil geometry measurement: (**a**) width measurement; (**b**) thickness measurement at three points.

**Figure 9 sensors-20-07053-f009:**
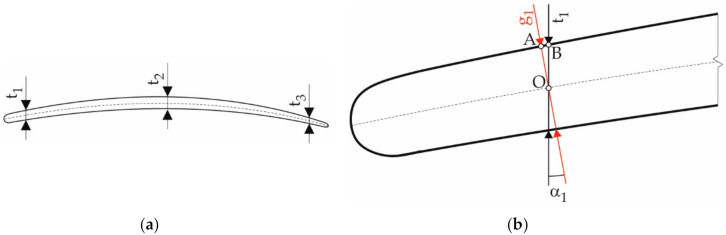
Geometric airfoil parameters: (**a**) parameters determined in the process of measurement; (**b**) corrected dimension due to the blade surface curvature.

**Table 1 sensors-20-07053-t001:** Measurement data.

Measurement Number	Width b [mm]	Thickness t_1_ [mm]	Thickness t_2_ [mm]	Thickness t_3_ [mm]
1	49.053	18.323	18.246	18.148
2	49.051	18.325	18.242	18.153
3	49.056	18.322	18.244	18.152
4	49.055	18.324	18.246	18.149
5	49.053	18.327	18.245	18.151
6	49.056	18.325	18.242	18.149
7	49.056	18.328	18.245	18.153
8	49.050	18.324	18.245	18.150
9	49.052	18.323	18.244	18.149
10	49048	18.327	18.242	18.148
11	49.055	18.323	18.246	18.151
12	49.053	18.324	18.244	18.149
13	49.055	18.323	18.245	18.152
14	49.048	18.325	18.244	18.149
15	49.051	18.322	18.245	18.153
16	49.050	18.326	18.243	18.148
17	49.056	18.324	18.246	18.151
18	49.054	18.325	18.245	18.149
19	49.049	18.322	18.244	18.148
20	49051	18.324	18.242	18.152
21	49.055	18.327	18.243	18.150
22	49.053	18.322	18.243	18.154
23	49.050	18.326	18.243	18.152
24	49.055	18.325	18.245	18.149
25	49057	18.327	18.245	18.151
26	49.055	18.322	18.244	18.152
27	49.056	18.328	18.245	18.150
28	49.053	18.325	18.243	18.148
29	49.049	18.323	18.244	18.152
30	49.055	18.326	18.246	18.153
dimensions of the reference element	49.060	18.330	18.251	18.157

**Table 2 sensors-20-07053-t002:** Analysis results.

	Width b	Thickness t_1_	Thickness t_2_	Thickness t_3_
mean $\bar{x}$	49.053 mm	18.325 mm	18.244 mm	18.151 mm
range r	0.009 mm	0.006 mm	0.004 mm	0.006 mm
accuracy A	0.007 mm	0.005 mm	0.007 mm	0.006 mm
repeatability EV	0.015 mm	0.010 mm	0.007 mm	0.010 mm
relative repeatability %EV	7.53%	5.02%	3.35%	5.02%
type A standard uncertainty	0.00049 mm	0.00034 mm	0.00023 mm	0.00034 mm

**Table 3 sensors-20-07053-t003:** Results of airfoil geometry measurement.

Cross-Section Number	Width b [mm]	Thickness t_1_ [mm]	Thickness t_2_ [mm]	Thickness t_3_ [mm]
1	33.643	1.146	1.607	0.587
2	33.277	1.189	1.729	0.524
3	32.993	1.330	1.984	0.664

**Table 4 sensors-20-07053-t004:** Angles determining the airfoil geometry.

Cross-Section Number	$\mathrm{Angle}\;\alpha_{1}\;[\mathrm{rad}]$	$\mathrm{Angle}\;\alpha_{2}\;[\mathrm{rad}]$	$\mathrm{Angle}\;\alpha_{3}\;[\mathrm{rad}]$
1	0.180	0.017	0.223
2	0.174	0.009	0.201
3	0.169	0	0.174

**Table 5 sensors-20-07053-t005:** Corrected airfoil dimensions.

Cross-Section Number	Width b [mm]	Thickness g_1_ [mm]	Thickness g_2_ [mm]	Thickness g_3_ [mm]
1	33.643	1.127	1.607	0.572
2	33.277	1.171	1.729	0.513
3	32.993	1.311	1.984	0.654

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
