# Peer review of "Device for Contact Measurement of Turbine Blade Geometry in Robotic Grinding Process"

_sensors, 2020, doi:10.3390/s20247053_

Round 1

Reviewer 1 Report

The article discusses the design, implementation, and testing of the accuracy of a measuring device used to measure the thickness of aircraft engine blades subjected to a robotic grinding process. Due to the small size and narrow tolerances of the blades, the presented device uses a contact measurement method. However, the proposed device used 8 commercial contact sensors (GT2-A32 sensors (Keyence, Osaka, Japan)) to measure the workpiece. I do not see obvious difference and novel principle for this case and other existing technologies. Its contribution is not enough. Therefore I think this manuscript should not be published.

Reviewer 2 Report

The article describes the development of device for contact measurement of blades during their production. The article is well designed and structured. The text clearly describes what the goal of the author’s team is and how it was achieved (will be achieved). I have few recommendation notes:

Line 39: The authors state as a disadvantage of laser solutions their relative inaccuracy, while stating that the requirement for measuring accuracy of a measuring instrument must be at least 0.01mm. However, current laser solutions have such accuracies - even up to 0.001 mm. Please edit or supplement this statement in the text.

Figure 2: Pleas correct the word “airofoil” to airfoil in the left bottom corner of the Figure 2.

Line 188: If the same operator is considered, then the way in which the blades are gripped by the operator / robot should be discussed and the error or repeatability of this procedure described. Or is it intentionally not yet considered?

Line 190: It would be appropriate to give an idea of how long the actual measuring process takes and how long it takes to measure all the individual required sections on the blade. This can be an important indicator of why to choose the technology you are developing over the available ones. Please add this information to the text.

Line 194: Table 1 shows an example of 10 measured values. If the measurement process is automated (or semi-automated), it would be appropriate to base the statistics on more than just 10 values so that the results are reliable.

Line 237: How is the alpha angle used to correct the measurement obtained? Is it a measured value or a value read from a drawing? Please add this information to the text.

Reviewer 3 Report

This manuscript presents device for measuring turbine blade geometry in robotic grinding process. Turbine blades are considered one of the most difficult products in terms of measuring geometric characteristics. Therefore, the topic of the manuscript is relevant for the subject area of metrology.

The manuscript proposes a device for contact measurement of the thickness and width of a blade in three sections using eight sensors. Typically, the blade profile is measured in three sections. Then a large number of parameters are calculated. For a robotic grinding process, it is enough to restrict only to the thickness, since it will affect the machined allowance.

Data on the error of the device are given. The experimental data prove that the proposed device has the necessary metrological characteristics.

The manuscript has structured in a logical way, including a problem definition, description of the mathematical model, analysis of simulation, discussion, and references used.

These research results are original and have scientific value. Nevertheless, there are certain moments demanding explanations.

  1. The manuscript does not indicate how the turbine blade is fixed on the machine tool and in the measuring device. Is the blade fixing scheme the same when processing and measuring?
  2. It is not clear from the text and figures how the sensors are configured to measure in the normal direction at a given surface point.
  3. The formula for recalculating the blade thickness, taking into account the angle of the normal to the profile, seems too rough.

4. The absolute and relative repeatability are given as a measure of the instrument error estimation. It is advisable to use the uncertainty of measurement according to ISO / IEC GUIDE 98-3: 2008.

Round 2

Reviewer 1 Report

The manuscript has been revised according to the reviewers' comments. Therefore, it can be accepted now.